# A Tension/Pressure Integrated Resistive Sensor Comprising of a PDMS-LC-MWCNT Composite

**DOI:** 10.3390/s21186078

**Published:** 2021-09-10

**Authors:** Miao Luo, Yumeng Zhang, Yuxiang Luo, Jiangang Lu

**Affiliations:** National Engineering Lab for TFT-LCD Materials and Technologies, Department of Electronic Engineering, Shanghai Jiao Tong University, Shanghai 200240, China; luomiao@sjtu.edu.cn (M.L.); zhangyumeng0324@sjtu.edu.cn (Y.Z.); sjtu_lyx@sjtu.edu.cn (Y.L.)

**Keywords:** tension/pressure integrated sensor, flexible resistive sensor, MWCNT-LC-PDMS composite film, hydrophobicity

## Abstract

A flexible strain sensor which integrates both pressure sensing and tension sensing functions is demonstrated with an active layer comprising of polydimethy-lsiloxane (PDMS) elastomer, liquid crystal (LC), and multi-walled carbon nanotubes (MWCNTs). The introduction of LC improves the agglomeration of MWCNTs in PDMS and decreases Young’s modulus of flexible resistive sensors. The tension/pressure integrated resistive sensor not only shows a broad tensile sensing range of 140% strain but also shows a good sensitivity of the gauge factor, 40, with tensile force. Besides, the tension/pressure integrated resistive sensor also shows good linearity and sensitivity under pressure. The resistance of the pressure sensor increases as the applied pressure increases because of the decrease in the cross-sectional area of the path. The sensor also shows good hydrophobic properties which may help it to work under complex environment. The tension/pressure integrated sensor shows great promising applications in electronic skins and wearable devices.

## 1. Introduction

In recent years, the flexible strain sensors are widely applied in health monitoring, wearable devices and electronic skins [1,2,3]. Various conductive nanomaterials (e.g., graphene, multi-walled carbon nanotubes (MWCNTs) and carbon black (CB) [4,5,6]) and supporting materials (e.g., polydimethy-lsiloxane (PDMS), PU foam, and Ecoflex [7]) have been utilized for flexible sensors. Flexible tensile sensors which convert the mechanical deformation into electrical signals are researched for high stretchability and flexibility and easily accessible systems [8,9,10]. Because of their fast response time, easy accessible mechanism and simple structure, the resistive pressure sensors are investigated intensively which convert the applied pressure to the changes in resistance [11,12]. Introducing microstructures into the flexible sensor’s active layer not only decreases Young’s modulus of flexible resistive sensors but also raises the relative resistive change in the conducting region, which helps to form the high sensitivity [13,14]. The resistance of the traditional sensor decreased as the applied pressure increased due to the reduction in conductive path length [14]. Nevertheless, a great amount of flexible strain sensors may only detect one certain mechanical strain, the tensile force and pressure integrated sensing is rarely investigated. A different scheme of pressure sensor whose resistance increases as the applied pressure increases because of the decrease in the cross-sectional area of the path is rarely researched.

A flexible strain sensor which integrates both pressure sensing and tension sensing functions is proposed with an active layer comprising of liquid crystal (LC), MWCNT, and PDMS elastomer. The introduction of LC improves the agglomeration of MWCNTs in PDMS and forms the porous conductive polymer with LC droplets, which improves the performance of the sensor. A flexible strain sensor is fabricated with a PDMS-LC-MWCNT composite film, which shows wide tensile sensing range and good sensitivity with tensile force, good linearity, and sensitivity under pressure. The resistance of the pressure sensor increases as the applied pressure increases because a different scheme is adopted through which the cross-sectional area of the path influences the resistance of the sensor.

## 2. Materials and Methods

The active layer materials of the tension/pressure integrated resistive sensor are composed of LC, MWCNT, and PDMS. PDMS is an exciting material possibility for its facile processability, chemical inertness, and good biocompatibility. To balance mechanical strength and flexibility, the ratio of PDMS to curing agent is 10:1. The MWCNTs are used as conductive filling materials for not only the good conductivity but also the good toughness. A low-polarity LC, TS029, is introduced to form the porous conductive polymer with LC droplets and improve the agglomeration of MWCNTs in PDMS since the end-groups of the LC molecules possess fantastically weak dipolar moments which are similar to PDMS.

Figure 1 shows the infrared spectrometric results of LC, MWCNT, and LC-MWCNT. In the IR spectrum of MWCNT, the most important band at 1637 ${\mathrm{cm}}^{-1}$ is related to C=C. In addition, the band at 3445${\;\mathrm{cm}}^{-1}$ refers to a few acidic sites on the surface of MWCNT. It can be seen from the curve of LC-MWCNT composite material that, in addition to functional groups on the MWCNT, there is also the characteristic absorption peak of LC molecular groups. Furthermore, 2919${\;\mathrm{cm}}^{-1}$ is the characteristic absorption peak of C-H bond of -CH3, and 2956${\;\mathrm{cm}}^{-1}$ and 1446${\;\mathrm{cm}}^{-1}$ are the characteristic absorption peaks of C=C of benzene ring. No new group characteristic peaks indicate that the LC interacts with MWCNTs via a non-covalent π−π interaction. Figure 2 shows the morphology of the MWCNT-LC-PDMS composite with the concentration of LC in PDMS from 0 wt% to 60 wt% with the same magnification under a polarized optical microscope (POM, XPL-30TF, Shanghai WeiTu Optics & Technology Co., Ltd., Shanghai, China). The surface of the composite becomes relatively flat and the agglomeration of the MWCNTs is reduced when the LC’s concentration increases from 0 wt% to 60 wt%, as shown in Figure 2. After the functionalization of LC, the MWCNTs are relatively uniformly dispersed in the PDMS and the miscibility of MWCNTs and PDMS is effectively improved.

To investigate the performance of the sensor with different concentration of composite, the concentration of MWCNT in PDMS is set from 8 wt% to 12 wt% with a step of 2 wt%, and the concentration of LC in PDMS is set from 0 wt% to 60 wt% with a step of 20 wt%.

Figure 3 shows the fabrication process of the sensor. At first, the materials of PDMS, LC, and MWCNT are mixed and stirred until the materials become homogeneous at room temperature. Then, the materials need to be removed the air through a vacuum chamber for about 15 min. We use a Teflon mold with a hollowed area of 40 mm ∗ 4 mm ∗ 1 mm from which the film can be easily peeled off. The PDMS-LC-MWCNTs mixture is poured into the Teflon mold and the surface of the PDMS-LC-MWCNTs mixture is smoothed through a scraper to keep the surface of the PDMS-LC-MWCNTs flat and the height of the PDMS-LC-MWCNT mixture same as the height of the Teflon mold. After that, the mold is annealed for about 3 h at 100 ℃ until the mixture is cured completely. Finally, the flexible film is peeled off from the Teflon mold and deposited on the copper electrodes.

## 3. Measurement and Discussion

Figure 4 shows the stretchability of the tension sensor. When the LC’s concentration raises from 0 wt% to 60 wt%, the elongation’s ratio for breaking of the sensor increases by a factor of 1.6 when the concentration of MWCNTs is 8 wt%. As the LC’s concentration changes from 0 wt% to 60 wt%, the elongation’s ratio for breaking of the sensor increases by a factor of 2.5 when the concentration of MWCNTs is 10 wt%. With the LC’s concentration increasing from 0 wt% to 60 wt%, the elongation’s ratio for breaking of the sensor increases by a factor of 3.41 when the concentration of MWCNTs is 12 wt%. The sensor with a higher concentration of MWCNT shows a higher increase in tensile sensing range compared with the sensor of relative lower concentration of MWCNT because the agglomeration of MWCNT in PDMS is significantly improved by the LC molecules. Meanwhile, the stretchability of the tension sensor is significantly improved as the introduction of LC into PDMS-MWCNT composite forms the porous conductive polymer with LC droplets, resulting in the decrease in the Young’s modulus of the flexible resistive sensor’s active layer. The sensor may achieve a maximum tensile sensing range of 140% strain when the concentration of MWCNTs is 8 wt% and the concentration of LC is 60 wt%. Qin et al. propose the GR/CNTs-PDMS flexible strain sensor whose elastic recovery rate is 3.42% after stretching 100 times with a strain of 30% [15]. By the comparison, the tension/pressure integrated resistive sensor comprising of a PDMS-LC-MWCNT composite shows good elasticity and stretchability.

According to the fitting curve shown in Figure 5, the slope of the curve indicates the measurement of the flexible strain sensor’s sensitivity which is termed as a gauge factor (GF). The sensitivity decreases from 6.97 to 2.74 when the MWCNTs’ concentration raises from 8 wt% to 12 wt% without LC. When the LC’s concentration increases from 0 wt% to 60 wt%, all the sensors’ sensitivity decreases at first and then increases. The least square method of the curve fitting shows the linearity of the sensor. When the concentration of the MWCNTs is 8 wt%, the sensor’s linearity decreases as the LC’s concentration increases. When the LC’s concentration changes to 60 wt%, the resistance response curve shows two linear segments, the sensitivity of the GF, 10.9, with low tensile force of 0–60% strain and the sensitivity of the GF, 40, with high tensile force of 60–100% strain, as shown in Figure 6. When the concentration of MWCNTs is high, 12 wt%, the sensor shows a good linearity with different LC concentration. When the LC concentration increases to 60 wt%, the sensitivity is higher than that without LC and the stretchability is almost 2.5 times of the sensor without LC. The introduction of LC improves the agglomeration of MWCNTs and forms the porous conductive polymer with LC droplets, which extends the range of the resistance of the tension sensor and improves the flexible sensor’s sensitivity.

The hysteresis of the flexible sensor is described by Equation (1), where ∆Hmax denotes maximum hysteresis error and $Y_{\mathrm{FS}}$ denotes the maximum vertical value of the curve. Figure 7 shows the flexible strain sensor’s hysteresis decreases from 12.8% to 7.8% when the concentration of LC increases from 0 wt% to 60 wt%, and the concentration of MWCNTs is 8 wt%. The hysteresis of the flexible strain sensor decreases from 18.9% to 8.5% when the concentration of LC increases from 0 wt% to 60 wt% and the concentration of MWCNTs is 10 wt%. The hysteresis of the flexible strain sensor decreases from 25.8% to 10.3% when the concentration of LC increases from 0 wt% to 60 wt% and the concentration of MWCNTs is 12 wt%. As the active layer of the sensor is difficult to recover to its original status when the tensile force is released due to the agglomeration of MWCNTs, after introducing the LC into the composite, the compatibility between MWCNTs and PDMS and the agglomeration of MWCNTs are significantly improved, resulting in the decrease in hysteresis.
(1)$$r_{H}=\pm \frac{∆H_{\max}}{Y_{\mathrm{FS}}}*100\%$$

The resistance change in the flexible resistive sensors under pressure is measured, as shown in Figure 8. As the LC’s concentration raises from 0 wt% to 60 wt% and the MWCNTs’ concentration is 8 wt%, 10 wt%, and 12 wt%, respectively, the sensitivity of the flexible resistive sensor under pressure increases by a factor of 20, 12, and 9 times. Furthermore, the sensor may significantly improve the pressure sensitivity only with the high LC concentration of 60 wt%. As the LC’s concentration of the flexible sensor increases forming a higher density of LC droplets, the Young’s modulus of the flexible sensor’s active layer deceases rapidly and the conductive path between two electrodes is easier to be cut off with the increased pressure, which improves the sensitivity of the flexible sensor. Meanwhile, the relative low concentration of MWCNTs of the flexible resistive sensor shows high pressure sensitivity because the conductive paths are difficult to cut off with more MWCNTs in the active layer.

Since the pressure sensor with low concentration of MWCNTs showed a high sensitivity, the response time was measured when the LC’s concentration in the flexible resistive sensor changes from 0 wt% to 60 wt% and the MWCNTs’ concentration is set to 8 wt%. As shown in Figure 9, while a pressure of 15 N is applied on the sensor and removed later, the response time is 0.21 s and the recovery time is 0.39 s, with the LC’s concentration of 0 wt%. The response time shows 0.15 s and the recovery time shows 0.33 s, with the LC’s concentration of 20 wt%. The response time is 0.18 s and the recovery time is 0.19 s, with the LC concentration of 40 wt%. Finally, the response time shows 0.10 s and the recovery time shows 0.14 s, with an LC concentration of 60 wt%. Qin et al. propose the GR/CNTs-PDMS flexible strain sensor whose response time is about 0.15 s [15], which is similar to the sensor we propose. The response and recovery time of the sensors, which have different LC concentrations are all in the sub-second level. With the increase in the LC concentration, the stability of on and off states are significantly improved, and both the response time and the recovery time become faster.

Figure 10 shows the measurement result of water contact angle of the flexible sensor. As discussed in the Wenzel models which is stated by Equation (2), the films’ water contact angle is closely related to the surface roughness. Where θ represents the water droplet’s contact angle on the surface of the rough film, r represents the roughness of the film’s surface, and $\theta_{0}$ is the water contact angle of the ideal MWCNT-LC-PDMS composite.
(2)$$\cos\theta \;={\;\mathrm{rcos}\theta }_{0}$$

The water contact angle of the surface of the flexible resistive sensor raises with the increase in the MWCNTs’ concentration, because the exposure of the MWCNTs on the surface of the flexible resistive sensor increases the surface roughness of the PDMS-LC-MWCNT film. With the increase in LC concentration, the water contact angle increases at first and then decreases. With the introduction of LC, the hydrophobic property of LC increases the water contact angle of the ideal MWCNTs-LC-PDMS composite and improves the hydrophobic property of the flexible resistive sensor. However, if the LC concentration continues to increase, the film surface roughness will decrease, as shown in Figure 2, which may degrade the hydrophobic property of the flexible sensor. The contact angle may reach 131° when the concentration of MWCNTs is 12 wt% and the concentration of the LC 20 wt%, which shows good hydrophobic property.

## 4. Conclusions

A flexible strain sensor, which integrates both the pressure and tension sensing functions, is demonstrated with an active layer comprising of LC, MWCNT, and PDMS. Thus, the introduction of LC not only improves the agglomeration of MWCNTs in PDMS but also reduces the Young’s modulus of the flexible resistive sensor. The sensor not only shows a broad tensile sensing range of 140% strain but also shows a great sensitivity of the gauge factor, 40, with tensile force. It also shows good linearity and sensitivity under pressure. The sensor also shows good hydrophobic property which may help it to work under complex environment. The tension/pressure integrated sensor shows great promising applications in electronic skins and wearable devices.

## Figures and Tables

**Figure 1 sensors-21-06078-f001:**
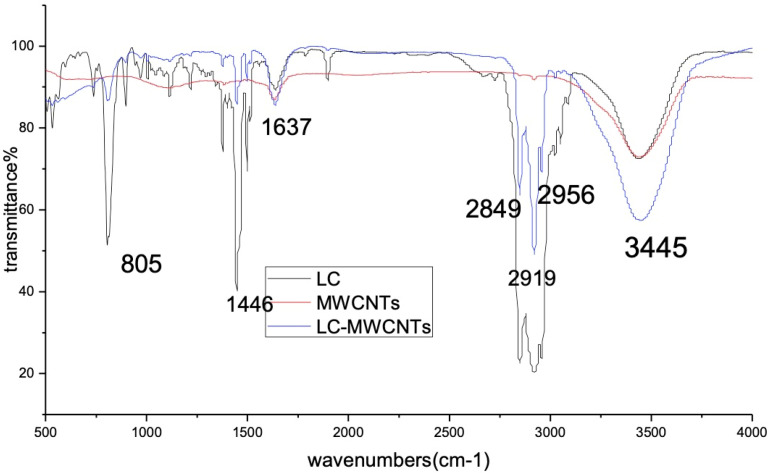
The infrared spectrometric results of MWCNTs, LC and MWCNTs-LC.

**Figure 2 sensors-21-06078-f002:**
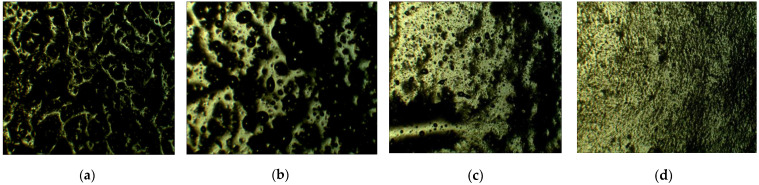
Morphology of the multi-walled carbon nanotubes-liquid crystal-polydimethylsiloxane (MWCNT-LC-PDMS) composite with concentrations of LC in PDMS of (**a**) 0 wt%; (**b**) 20 wt%; (**c**) 40 wt%; and (**d**) 60 wt%.

**Figure 3 sensors-21-06078-f003:**
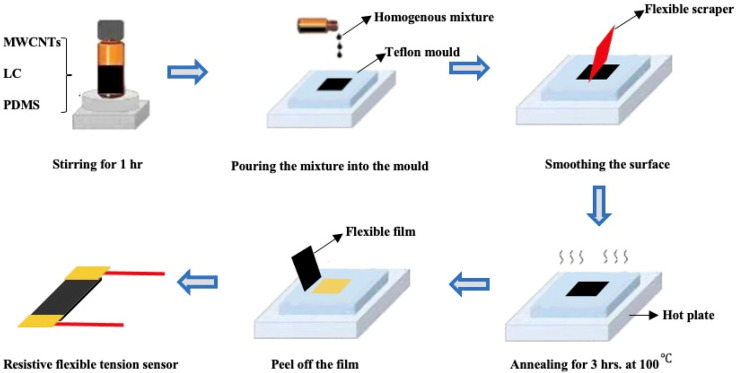
The fabrication process of the flexible resistive sensor.

**Figure 4 sensors-21-06078-f004:**
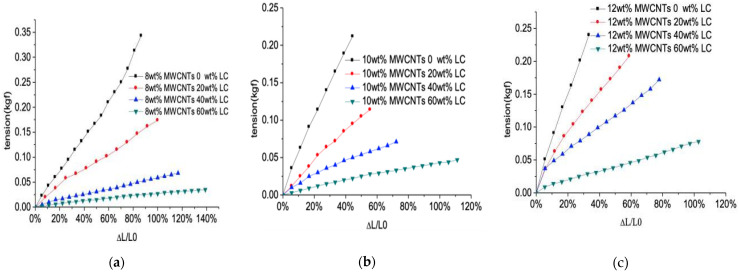
The strain of the tensile sensor with concentration of MWCNT of (**a**) 8 wt%; (**b**) 10 wt%; and (**c**) 12 wt%.

**Figure 5 sensors-21-06078-f005:**
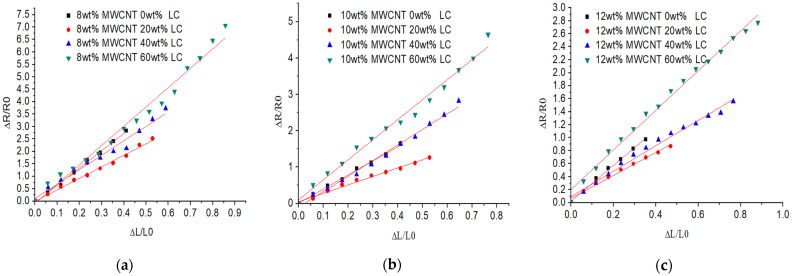
The relative resistance changes under different strain of the tensile sensor with 0–60 wt% LC and MWCNTs of (**a**) 8 wt%; (**b**) 10 wt%; and (**c**) 12 wt%.

**Figure 6 sensors-21-06078-f006:**
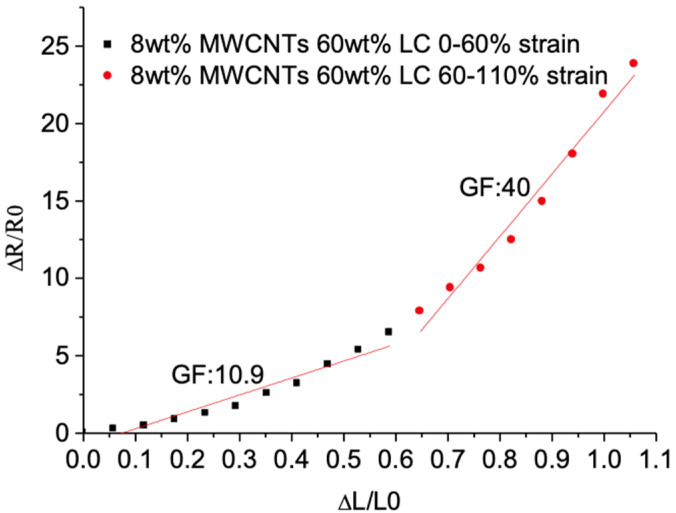
The relative resistance changes under different strain of the tensile sensor with 8 wt% MWCNTs and 60 wt% LC.

**Figure 7 sensors-21-06078-f007:**
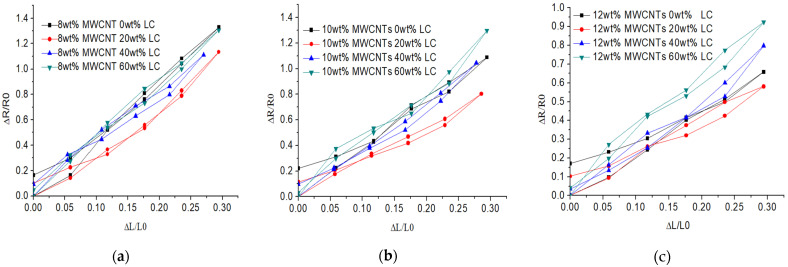
The hysteresis of the tensile sensing with 0–60 wt% LC and MWCNTs of (**a**) 8 wt%; (**b**) 10 wt%; and (**c**) 12 wt%.

**Figure 8 sensors-21-06078-f008:**
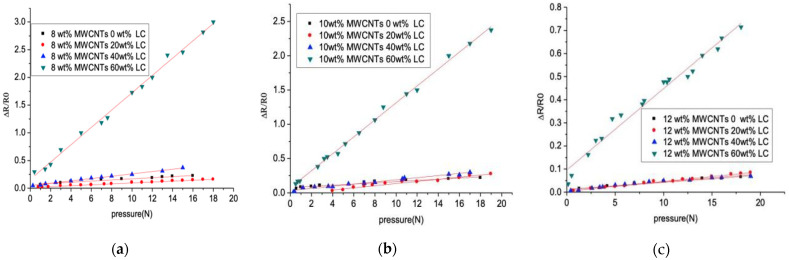
The relative resistance changes in the pressure range of 0–20 N with 0–60 wt% LC and MWCNTs of (**a**) 8 wt%; (**b**) 10 wt%; and (**c**) 12 wt%.

**Figure 9 sensors-21-06078-f009:**
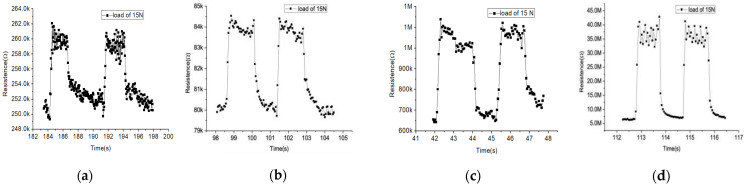
Response and recovery time of the pressure sensor with 8 wt% MWCNTs and LC of (**a**) 0 wt%; (**b**) 20 wt%; (**c**) 40 wt%; and (**d**) 60 wt%.

**Figure 10 sensors-21-06078-f010:**
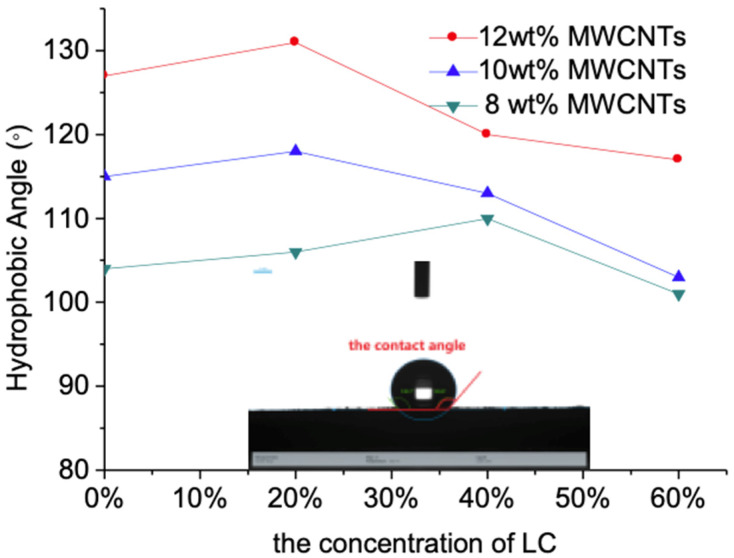
The measurement of water contact angle of the flexible sensor.

## Data Availability

The data presented in this study are available on request from the corresponding author.

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
