# Peer review of "A Tension/Pressure Integrated Resistive Sensor Comprising of a PDMS-LC-MWCNT Composite"

_sensors, 2021, doi:10.3390/s21186078_

Round 1
Reviewer 1 Report
The authors proposed a flexible strain sensor of pressure sensing and tension sensing functions. It has a broad tensile sensing range strain, good sensitivity of the gauge factor, good linearity and sensitivity under pressure, and good hydrophobic property. This work is very interesting in field of electronic skins and wearable devices. Therefore, I recommend it for publication after the following the following minor revision.
- The resistance change of the flexible resistive sensor under pressure is different from the traditional pressure resistive sensors, please describe the characteristics of the flexible resistive sensor compared with the traditional pressure resistive sensors.
- Please add more recent references and pay attention to the format of the references.
Author Response
Please see the attaachment

Reviewer 2 Report
Please find the attachment.

Reviewer 3 Report
The authors should clearly explain in Abstract and Introduction what are the novel findings added in this paper in comparison with their earlier work. In the previous work (ref. [15]), some results and conclusions were already presented for the same composite.
Figure 2 is not adequately described in the caption. It should explain the kind of the obtained images (e.g. 'microphotographs') instead of using the unclear term 'surface measurements'. Also, the scale indicator should be included in the images.
References should be formatted consistently - some references are given with article titles, others without titles etc.
For a more complete context of the discussion, authors could compare the presented results with similar sensors studied by other research institutions.
Round 2
Reviewer 2 Report
The manuscript entitled “A Tension/Pressure Integrated Resistive Sensor comprising of a PDMS-LC-MWCNT Composite”
The authors amended the manuscript according to the referee’s comments. In addition, they have explained why the MWCNT is used in this study in the manuscript. Therefore, I recommend the paper can be published in Sensors.